# Enhanced pupillary light reflex in infancy is associated with autism diagnosis in toddlerhood

Pär Nyström[1], Teodora Gliga[2], Elisabeth Nilsson Jobs[1], Gustaf Gredebäck[1], Tony Charman [3], Mark H. Johnson[2,4], Sven Bölte[5,6] & Terje Falck-Ytter[1,5,6,7]

Autism spectrum disorder (ASD) is a neurodevelopmental condition affecting around 1% of the population. We previously discovered that infant siblings of children with ASD had stronger pupillary light reflexes compared to low-risk infants, a result which contrasts sharply with the weak pupillary light reflex typically seen in both children and adults with ASD. Here, we show that on average the relative constriction of the pupillary light reflex is larger in 9–10-month-old high risk infant siblings who receive an ASD diagnosis at 36 months, compared both to those who do not and to low-risk controls. We also found that the magnitude of the pupillary light reflex in infancy is associated with symptom severity at follow-up. This study indicates an important role of sensory atypicalities in the etiology of ASD, and suggests that pupillometry, if further developed and refined, could facilitate risk assessment in infants.

[1] Uppsala Child & Babylab, Department of Psychology, Uppsala University, SE-75142 Uppsala, Sweden. [2] Centre for Brain and Cognitive Development, Birkbeck, University of London, London WC1E 7HX, UK. [3] Department of Psychology, Institute of Psychiatry, Psychology and Neuroscience, King's College London, London SE5 8AF, UK. [4] Department of Psychology, University of Cambridge, Cambridge CB2 3EB, UK. [5] Pediatric Neuropsychiatry Unit, Department of Women's and Children's Health, Center of Neurodevelopmental Disorders at Karolinska Institutet (KIND), Karolinska Institutet, SE-171 Stockholm, Sweden. [6] Child and Adolescent Psychiatry, Center for Psychiatry Research, Stockholm County Council, SE-11330 Stockholm, Sweden. [7] Swedish Collegium for Advanced Study (SCAS), SE-752 Uppsala, Sweden. Correspondence and requests for materials should be addressed to T.F-Y. (email: terje.falck-ytter@psyk.uu.se)

utism spectrum disorder (ASD) is a heterogeneous, heritable, and common condition characterized by early onset alterations in social communication and interaction alongside stereotyped behaviors, intense interests, and sensory atypicalities[1]. ASD is typically chronic, and associated with functional impairment, mental health issues, and low quality of life. Although the etiology of ASD is both complex and heterogeneous, genetic, and environmental factors are believed to converge in a limited number of biological pathways related to brain development during the first years of life[2,3]. Yet, the precise mechanisms involved remain uncertain. Moreover, there is no evidence-based pharmacological treatment convincingly demonstrating improvement of the core symptoms of ASD, and behavioral interventions are time consuming and typically of moderate effect[4]. Consequently, there is a strong need to identify new measures that can illuminate the underlying biological processes, assist early risk assessment, and facilitate clinical stratification and evaluation of treatment efficacy[5].

The pupillary light reflex (PLR) regulates the amount of light that reaches the retina, and the reflex pathway involves the ganglion cells of the retina, the Pretectal nucleus, the Edinger–Westphal nucleus, and the Cilary ganglia[6]. Several studies indicate an attenuated PLR in children and adults with an ASD diagnosis relative to controls[7,8]. In children with ASD, the relative constriction (but not the latency) of the reflex is correlated to the amount of their sensory atypicalities[9]. Surprisingly, we recently discovered that unlike children and adults with an ASD diagnosis[7,8], 10-month-old infants at high risk for ASD (due to having an older sibling with the disorder) had stronger PLRs than low-risk-control infants with no family history of ASD[10]. This finding raised the possibility that the PLR might not only associate with ASD, it might also describe neurodevelopmental processes that are on an atypical trajectory in infants that will develop this disorder. However, the previous study[10] did not include any outcome data, neither in terms of categorical diagnostic outcome nor dimensional measures of ASD traits, which is crucial in order to confirm the hypothesized link between enhanced PLRs in infancy and ASD.

Here, we followed up the infants in our previous study (the EASE sample)[10] to 3 years, an age when ASD can be diagnosed with high accuracy. To increase the sample size we also included a second dataset from the BASIS group (BASIS sample). Our combined sample of n = 187, with 29 ASD participants, offered increased power for testing our main hypothesis—that infants with later ASD have stronger PLRs than typically developing (TD) infants[10]. Unless otherwise specified, we will report the results based on the combined EASE and BASIS samples (see Supplementary Methods for site-specific information), using three groups: 1) high risk infants later diagnosed with ASD (HR-ASD); 2) high risk infants who did not receive an ASD diagnosis at follow up (HR-no-ASD); 3) a control group of TD infants. We also analyzed the pupil data longitudinally in infancy to see if we could find support for the presence of different developmental trajectories of the reflex in ASD vs. controls.

## Results

**PLR at 9–10 months in relation to later diagnosis.** The statistical analysis revealed a significant main effects of group in terms of the relative constriction of the PLR, $F(2, 184) = 6.4$, $P = 0.002$, $\eta p^2 = 0.065$. Planned comparisons between groups showed that the HR-ASD group differed from the other groups: HR-ASD vs. TD, $P = 0.001$, 95% CI (0.087 to 0.307); HR-ASD vs. HR-no-ASD, $P = 0.006$, 95% CI (0.039 to 0.226); HR-no-ASD vs. TD, $P = 0.122$, 95% CI (−0.018 to 0.0147). The following statistics represent the data as plotted in Fig. 1b: TD, $n = 40$, mean = 1.00, SD = 0.21; HR-no-ASD, $n = 118$, mean = 1.06, SD = 0.23; HR-ASD, $n = 29$, mean = 1.20, SD = 0.24. Adding the Mullen Scales of Early Learning (MSEL) 10 months total score, data quality (amount of missing data) or gender as covariates did not change these results (see Supplementary Note 1 and Supplementary Figs. 1-2 for details).

No significant effects of group on PLR latency were found: $F = (2, 184) = 2.268$, $P = 0.106$, $\eta p^2 = 0.024$, and planned comparisons showed no significant differences between any pair of groups. More details about the latency results, as well as an analysis of the baseline pupil size (which did not differ between groups) is found in Supplementary Note 2, and details for each site separately is found in Supplementary Note 3.

**PLR at 9–10 months in relation to later symptom severity.** Next, we performed Pearson correlations to investigate the

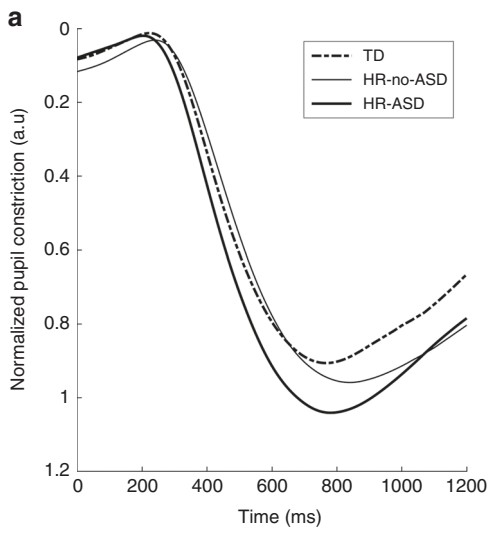

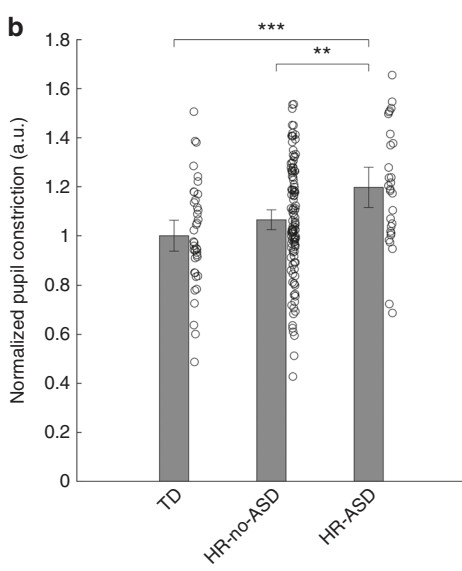

**Fig. 1** Stronger PLR in infancy is associated with ASD diagnosis at three years of age. **a** Average pupil traces for all infants expressed as the site-normalized relative constriction of the pupil following the stimuli onset at 0 ms. **b** Mean relative constriction normalized within site by dividing with the TD group average (see main text), together with individual data points. Error bars are 95% CI

relationship between Autism Diagnostic Interview-Revised (ADI-R) algorithm scores and Autism Diagnostic Observation Schedule-2 (ADOS-2) comparison scores (CS) and the PLR measures. We excluded the TD group in these dimensional analyses. No significant correlations were found for the latency measure, but there were significant correlations between ADI-R and relative constriction ($n = 142$, $r = 0.197$, $P = 0.019$), and between ADOS-2 CS and relative constriction ($n = 144$, $r = 0.196$, $P = 0.018$; Fig. 2a, b). We also assessed the relation between relative constriction and the two subscales of the ADOS-2, and found that relative constriction was significantly associated with the social affect comparison scores (SA CS) ($n = 144$, $r = 0.218$, $P = 0.009$), but not the restrictive repetitive behaviors (RRB) algorithm scores ($n = 144$, $r = 0.136$, $P = 0.103$; Fig. 2c, d). Site did not moderate these correlations. Including the TD group in the analysis (i.e., analyzing all children together) resulted in a similar result, except that the RRB scale now showed a trend in the same direction (ADI-R total $n = 175$, $r = 0.193$, $P = 0.010$; ADOS CS $n = 179$, $r = 0.216$, $P = 0.004$; ADOS SA CS $n = 179$,

$r = 0.230$, $P = 0.002$; ADOS RRB $n = 179$, $r = 0.140$, $P = 0.062$; Supplementary Fig. 3).

**Longitudinal analysis of the PLR.** Finally, we tested if group difference in pupil relative constriction changed over time[8,10]. This was possible because the infants had longitudinal pupil assessments available at 14 months (EASE sample) and at 15 months (BASIS sample). As above, the TD group's performance at the first time point was used for normalization for all groups/timepoints within sites. As a measure of change, we subtracted each infant's relative constriction measure at the first time point from that individual's relative constriction measure at the second time point. Values >0 indicated increased constriction over time and values <0 indicated decreased constriction (descriptive statistics: HR-ASD $n = 21$, mean = -0.057, SD = 0.214; HR-no-ASD $n = 93$, mean = 0.051, SD = 0.256; TD $n = 27$, mean = 0.102, SD = 0.248). Using this difference score as our dependent measure and group as fixed factor, we found a marginally significant main effect of group, $F(2, 138) = 2.51$, $P = 0.085$, $\eta p^2 = 0.035$ (this is analogous to an interaction effect using repeated measures ANOVA with a group factor and an age factor: the statistics are identical). Planned comparisons between the groups showed a significant difference between the HR-ASD group and the TD group, $P = 0.030$ (95% CI ($-0.303$ to $-0.016$)), and marginally between the HR-ASD and the HR-no-ASD groups, $P = 0.074$ (95% CI ($-0.228$ to 0.011)), but not between the HR-no-ASD and TD groups, $P = 0.352$ (95% CI ($-0.159$ to 0.057)). This longitudinal analysis supported the hypothesis of different developmental trajectories in terms of the relative constriction of the PLR constriction for the TD and HR-ASD group. This finding is generally in line with the possibility that group differences in the PLR reverse over time (see Supplementary Note 4 and Supplementary Fig. 4 for more descriptive statistics).

**Discussion**

The results of this study show that that the magnitude of constriction of the pupil in response to changes in light in infancy is associated with ASD diagnosis at 3 years of age. The HR-ASD group differed both from the HR-no-ASD group and the TD group. The HR-no-ASD group is at elevated risk for a range of developmental difficulties such as cognitive and motor delays, language problems, and difficulties with attention and hyperactivity;[11] hence it is a clinically relevant comparison group. That the group differences were paralleled by the results of the dimensional analysis (Fig. 2) suggests that the PLR is associated with later ASD symptom severity, not only categorically defined ASD. This finding is consistent with the view that ASD should be seen as the extreme end of a neurodevelopmental continuum encompassing the whole population[12] (see also Supplementary Fig. 3).

An atypical PLR appears more likely to index general atypicalities in brain development than selective disruptions of the so-called 'social brain' (for discussion, see ref. [13]). More

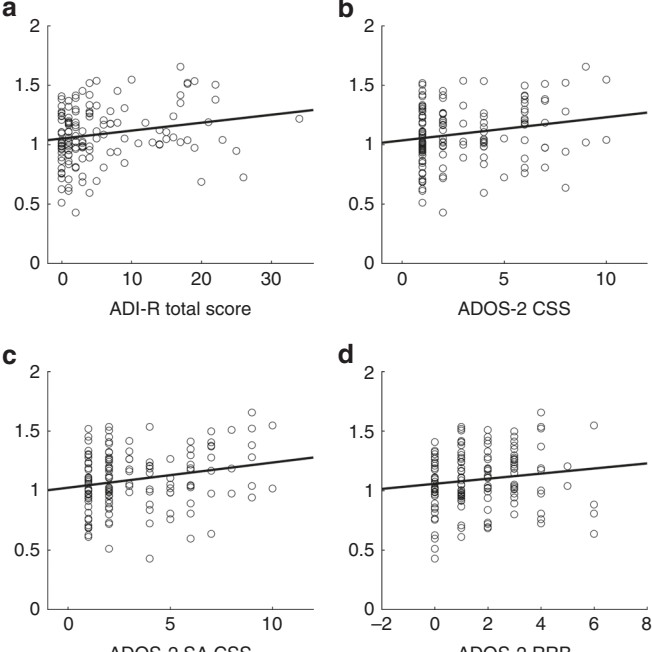

**Fig. 2** Scatterplots showing the association between normalized relative constriction of the PLR in infancy and severity measures of ASD symptoms at three years of age. Large relative constriction of the PLR in infancy was associated with having higher **a** ADI-R scores, **b** ADOS-2 comparison scores, and **c** Social affect (SA CS) comparison scores (ADOS-2 subscale). The relation for the Restricted and Repetitive Behaviors (RRB) algorithm scores (ADOS-2 subscale) did not reach statistical significance (**d**). Statistics in main text

**Table 1 Participant characteristics by group at the 10-month assessment, final samples (Mean/SD)**

| Measure | HR-ASD ($n = 29$, 7 girls) | HR-no-ASD ($n = 118$, 62 girls) | TD ($n = 40$, 20 girls) | GLM with group as fixed factor |
|---|---|---|---|---|
| Age (days) | 288.31/31.65 | 283.01/26.80 | 294.05/25.40 | $F(2, 184) = 2.532$ $P = 0.082$ |
| MSEL TOT[a] | 101.41/15.50 | 106.58/15.02 | 108.35/14.16 | $F(2, 184) = 1.947$ $P = 0.146$ |
| SES[b] | 3.21/1.35 | 3.79/1.21 ($n = 117$) | 4.38/1.00 | $F(2, 183) = 8.219$ $P < 0.001$ |

[a] Mullen Scales of Early Learning total score
[b] Socio-economic status based on parental education on a five level rank scale. While there is a significant difference between groups, this difference does not influence our main results: adding SES as a factor in the analysis did not change the pattern of results

specifically, because the PLR relative constriction reliably tracks the amount of sensory atypicalities in older children with ASD[9], our finding suggests that atypicalities in neural systems involved in sensory processing could play an important role in the early development of ASD[3,14]. We found significant prospective correlations for the SA (social affect) domain of the ADOS-2, but it should be noted that a (non-significant) trend in the same direction was also found for the behavioral flexibility (RRB) domain. Although more research is needed to clarify these relations, our results are generally in line with the idea that early emerging atypicalities in sensory processing constitute a unitary explanation for multiple symptom domains in ASD.

It has been hypothesized that because the PLR pathway is acetylcholine dependent, PLR atypicalities in ASD could reflect the disruptions of the cholinergic system[10,15–17], which plays a key role in the regulation of excitatory–inhibitory balance early in life[18,19]. A recent small sample study indicated that maternal phosphatidylcholine treatment altered sensory processing (prepulse inhibition) in infancy and prevented social withdrawal in early childhood via activation of the a7-nicotinic acetylcholine receptor[19]—a receptor type which mediates the PLR in some animal species[20]. However, cholinergic disruption is not the only possible explanation for the results. For example, one study indicated that the reflex is mediated in part by N-methyl-D-aspartic acid (NMDA) receptors in humans[21], a receptor type that has been linked to ASD in several studies (e.g., ref. [22]). Top down influences, originating outside the reflex arc itself, could also contribute to the observed effects[23,24]. Most likely, a combination of animal, molecular genetic, and human behavior/imaging research is best suited to shine light on the exact mechanisms driving the atypicalities observed here[5].

Our result supports the idea that group differences in PLR parameters may change over time (Supplementary Fig. 4). Although we did not observe a crossing over within our samples, we did observe a significant difference between the change scores in the TD and the HR-ASD groups, which is rather striking given that this analysis was based on data spanning only a few months in infancy. These findings are theoretically important, as any explanation for the result would need to encompass the developmental unfolding of these processes. These results are also encouraging from a clinical point of view, as they suggest that group differences may be even more pronounced earlier in development which could facilitate early detection.

The study has some important limitations. First, the results are based on pupil data from two sites using a non-identical methodology. Future studies would benefit from including data from a larger number of infants being tested under similar conditions. However, the use of different paradigms to elicit the PLR is also a strong point of this research, since it demonstrates that this finding is robust to experimental changes. PLR was opportunistically measured in the BASIS sample. This will hopefully encourage other groups to take a similar approach to existing data and replicate our findings. Second, the combination of data from two sites can be done in different ways, and we used the TD group as basis for normalization across sites. This builds on the assumption that the characteristics of the TD group are similar in both sites. However, we do show that the overall pattern of result remains when using any of the other groups as a basis for normalization (Supplementary Note 1). Although the results need to be interpreted with some caution due to these constraints, the findings nevertheless provide strong initial evidence that early pupillary atypicalities are related to later neurodevelopmental problems.

In sum, this study indicates that the pupil provides a unique and previously unrecognized window into the neural development of human infants, and could contribute to early detection of ASD. Other and larger studies of infants at risk are ongoing; these studies may illuminate whether the PLR is useful for defining sub-groups within the ASD population[5].

## Methods

**Subjects and procedures.** The study was approved by the Regional Ethical Board in Stockholm (EASE sample) and UK National Health Service, National Research Ethics Service London REC 08/H0718/76 (BASIS sample). Parents provided written informed consent. The study was conducted in accordance with the 1964 Declaration of Helsinki.

A total of 208 9–10-month-old participated in the experiments at the two sites. High risk infants with an older sibling with ASD ($n = 163$, 77 girls) (HR infants) were recruited from the projects' websites, advertisements, and via clinical units. Low-risk-control infants with typical development (TD infants) with no family history (up to second degree) of ASD ($n = 45$, 21 girls) were recruited from birth records or advertisements. All infants were born full-term ($\geq 36$ weeks), and did not have any confirmed or suspected medical conditions, including visual/auditory impairments. Of the initial sample of 208 infants, 187 contributed data to the final analyses (Table 1; see Supplementary Methods for details).

At 36 months, we collected standardized information on medical history, current developmental, and adaptive level, as well as autistic symptoms using the ADI-R[25], the ADOS-2[26], the Vineland Adaptive Behavior Scales[27], and MSEL[28]. The clinical evaluation was conducted without blindness to risk group by experienced clinical researchers (psychologists) with demonstrated research-level reliability. Based on the information, final DSM-5 judgements were made by a senior clinical researcher, and participants were assigned either to the ASD group, a high-risk without ASD group or a low-risk control group (Table 1).

In the EASE sample, pupil data were collected on a Tobii 1750 eye tracker (Tobii Technology, Danderyd, Sweden) with a sampling rate of 50 Hz in a room with a controlled ambient light level of 0.9 lux. The stimulus lasted ~6 s and consisted of a small central fixation point on a black background (0.9 lux) that flashed white (190 lux) for 75 ms with a random onset between 1600 and 2400 ms. The stimulus was presented 16 times to each infant. In the BASIS sample, pupil data were collected on a Tobii T120 eye tracker (Tobii Technology, Danderyd, Sweden) with a sampling rate of 60 Hz. The stimulus was presented 32 times to the participants and consisted in a small 1 s animation presented on a white background, followed by a black screen with a mean duration of 313 ms (SD = 140 ms), followed by another white background stimulus, lasting for 1.5 s(the same stimuli used in ref. [29]), which induced a reliable PLR response.

**Data analysis.** In line with previous research, the PLR was evaluated both in terms of its relative constriction and its latency[7,8,10]. The relative pupil constriction was calculated as in Fan et al.[7] by the formula $(A_0^2 - A_m^2)/A_0^2$, where $A_0$ is the average pupil diameter before onset of the PLR (during an interval starting 100 ms before and ending at the PLR onset, as determined by the PLR latency) and $A_m$ is the minimum pupil diameter in the interval 500–1500 ms relative to the stimulus onset (Fig. 1a, b, Supplementary Methods). In accordance with previous work[10,30], the PLR latency was defined at the acceleration minimum between flash onset and minimum pupil diameter for each trial. All PLR response values were normalized through division by the average of the TD group in each site separately; i.e., values >1 are higher than the TD average and values <1 are lower (see Supplementary Note 1 for more details of the transformation and raw data from both sites). This approach was used to account for differences in experimental settings between the EASE and BASIS cohort. All statistical tests were performed in SPSS (IBM SPSS Statistics for Windows, Version 24.0) using two-tailed ($\alpha = 0.05$) general linear models (GLMs) with group as fixed factor. Preliminary models tested whether site (EASE/BASIS) or baseline pupil size (as covariate) interacted with group status, but because these variables did not moderate the group differences, neither for the relative constriction measure (group X site $F(2, 178) = 1.945$, $P = 0.146$; group X baseline $F(2, 178) = 0.512$, $P = 0.600$) nor for the latency measure (group X site $F(2, 178) = 2.027$, $P = 0.135$; group X baseline $F(2, 178) = 1.798$, $P = 0.169$), the site and baseline factors were therefore dropped from the final models. All measures met the assumption of equal error variances as assessed by the Levene's test unless otherwise stated, and the normality of model residual distributions were confirmed using Q–Q plots. The mean value from all trials for each subject was used as the dependent variable.

**Data availability.** The PLR analysis workflow was implemented in the TimeStudio framework[31] and is publicly available with sensitive information removed (uwid=ts-100-71f) through the TimeStudio software. Relevant data are available from the corresponding author on a reasonable request.

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

## Acknowledgements

The authors thank the children and families participating in this research. We would also like to thank the EASE team (Sweden): Aschermann, S., Andersson Konke, L., Brocki, K., Cauvet, E., Lundin Kleberg, J., Ristolainen Spak, J., Thorup, E., Zander, E., and the BASIS team (UK): Baron-Cohen, S., Bedford, R., Bolton, P., Blasi, A., Cheung, C., Davies, K., Elsabbagh, M., Fernandes, J., Gammer, I., Green, J., Guiraud, J., Liew, M., Lloyd-Fox, S., Maris, H., O'Hara, L., Pasco, G., Pickles, A., Ribeiro, H., Salomone, E., Tucker, L., and Yemane, F. This study was supported by the Swedish Research Council in partnership with FAS, FORMAS, and VINNOVA (Crossdisciplinary Research Programme Concerning Children's and Young People's Mental Health, Grant Number 259-2012-24). T.F. Y. was supported by Stiftelsen Riksbankens Jubileumsfond (NHS14-1802:1), the Swedish Research Council (2015–03670), EU (MSC ITN 642996), and The Swedish Collegium for Advanced Study (SCAS/Pro Futura). S.B. was supported by the Swedish Research Council (523-2009-7054). G.G. was supported by the Wallenberg Foundation (KWA 2012. 0120). T.G., T.C., M.H.J., and BASIS were supported by MRC Programme Grant no. G0701484, and a funding consortium led by Autistica. The research leading to these results has received support from the Innovative Medicines Initiative Joint Undertaking under grant agreement n° 115300, resources of which are composed of financial contribution from the European Union's Seventh Framework Programme (FP7/2007-2013) and EFPIA companies' in kind contribution.

## Author contributions

T.F.Y. conceived the study and drafted the manuscript with contributions from P.N., T.G., G.G., E.N.J., T.C., M.H.J., and S.B. P.N. developed the experimental design in EASE study and was responsible for data analysis with contributions from T.F.Y. and T.G. T.G., T.C., and M.H.J. developed the experimental design in BASIS study. E.N.J. and S.B. performed the clinical evaluations at follow-up for EASE sample and T.C. for BASIS sample. All authors have critically revised the manuscript and approved the final version. T.F.Y., P.N., and T.G. performed the final revision of the manuscript for intellectual content.

## Additional information

**Competing interests:** S.B. discloses that he has in the last 5 years acted as an author, consultant or lecturer for Shire, Medice, Roche, Eli Lilly, Prima Psychiatry, GLGroup, System Analytic, Ability Partner, Kompetento, Expo Medica, and Prophase. He receives royalties for text books and diagnostic tools from Huber/Hogrefe, Kohlhammer, and UTB. The remaining authors declare no competing interests.

