## [Peer Review File · Nature Communications]

Reviewers' comments:

Reviewer #1 (Remarks to the Author):

The manuscript reports stronger pupillary light reflex (PLR) in infants at-risk for autism relative to those at-risk who do not develop autism or typically developing controls. They report eye tracking data that has been recently published (Molecular Autism 2015), but the current manuscript presents a re-analysis of PLR data against diagnostic outcomes at 2- years of age. Previous studies have reported differences in visual processing in infants later diagnosed with ASD, e.g., Gliga et al 2014 Curr Bio; Shic et al BioPsych). Yet, this is the first clear demonstration that a fairly low-level visual reflex is associated with a later diagnosis of autism.

The authors base their hypothesis on prior studies that suggest attenuated PLR in older children and adults with ASD (where their prior work suggests the opposite pattern earlier in life). The manuscript and the conclusions would be much stronger if this attenuation is directly observed in the same sample over development (between 10 months and 2-4 years). This is especially important given the small samples size of diagnosed children. Therefore, the conclusions would be bolstered by the inclusion of PLR data on the same infants at 2-3 years. My concern here is that overt behavioural risk signs of autism (affecting social interaction) emerge rather clearly around 12-months of age, which is only two months after the age that a stronger PLR is reported. Also, further follow up to 3-years to examine stability of diagnosis would be needed, given that only a provisional diagnosis can be ascertained at 2-years.

Methods used to calculate the PLR appear sound. However, my knowledge of the Tobii system with infants is that raw data is often contaminated by a number of artefacts. It would be great if the authors can provide further demonstration that these artefacts cannot account for the observed group differences, e.g., analysis of baseline data to show it does not differ in the three groups.

In terms of interpretation, I agree with the author's conclusion that the observed differences may reflect genetic liability. The current manuscript brings us a few steps closer to understanding potential genetic and cellular mechanisms, given that the PLR is biologically well characterized.

In contrast, I don't agree with the authors' conclusion that they have identified a "developmentally sensitive predictive biomarker of ASD". I think this is a misinterpretation of what a biomarkers is and its potential clinical utility. It's important to note, however, that Nature has set a precedent

for publishing conclusions about eye tracking in infants that have much bigger news than scientific or clinical value (Klin & Jones 2014). Therefore, I hope that the authors can maintain their focus on advancing scientific discovery without the pressure to prematurely suggest clinical utility where there is none.

Reviewer #2 (Remarks to the Author):

This study used the high-risk infant sibling design to investigate whether differences in the pupillary light reflex (PLR) may be an early risk or diagnostic marker for Autism Spectrum Disorder (ASD). It builds on a previously published study (Nystroem et al, 2015, Mol Autism) by the same group, which reported that 10-month old infants with high familial risk for ASD had a stronger PLR than low-risk infants.

ASD can currently only be reliably clinically diagnosed from around the age of 24 months onwards. Therefore, the current report goes crucially beyond the previous study by now comparing high-risk infants who did vs. did not develop ASD on the PLR.

I found the manuscript to be well written and the study design very clear. I agree that the PLR is a very interesting candidate biomarker for ASD for the reasons that were given.

However, I also had several queries/ concerns:

1. The sample size of the HR-infants who developed ASD is quite small (N=6). This should be considered in the context of the considerable costs of the HR infant design, both due to its longitudinal nature and because only (very roughly) 1 in 5 children that are originally enrolled develop ASD. Therefore, I would treat the results as potentially very interesting but in my opinion at this point they do not provide “strong evidence” for the value of the PLR as a risk/ diagnostic marker.
2. As in many studies, the authors seem to conclude from their finding of a significant group difference that the measure could be promising risk or diagnostic biomarker. However, the aim of a risk or diagnostic biomarker is to predict, for a given individual, whether or not they likely develop or have ASD. This can't be done based on mean group differences but requires at least an assessment of the accuracy of the measure (i.e., its sensitivity/ specificity). Individual values are nicely plotted in Fig 1 c for relative constriction and latency. But from that it appears as if there was some overlap in scores between the three groups. For example, on relative constriction three out of the six ‘ASD positive’ children had scores within ? SDs of the TD range (but above the TD mean). Conversely, three of the TD children seem to have scores within or above the ‘ASD positive’ mean. More information would be useful that gives some estimate of how sensitive/ specific would be the PLR in predicting if a child develops or has ASD?

3. The authors mention in the introduction that while they observed (on average) stronger PLRs in high-risk infants, other previous studies with children and adults with ASD found weaker PLRs. They argue that therefore the PLR may not only predict ASD but also “reveal neural processes that are on a highly deviant developmental trajectory”. Again, I understand their point and agree that this would be an interesting possibility. However, the current study is not able to support/ speak to this; indeed there remains a possibility that differences in these study findings could be due to other factors that may be important to understand in order to assess the potential clinical biomarker utility of the PLR.

Reviewer #3 (Remarks to the Author):

The authors present follow up data on a study of high-risk infants who participated in a pupillary light reflex (PLR) experiment at 10 months of age. Between trials of point light displays, infants saw a 75ms white flash preceded and followed by a black screen. Using a video based eye-tracker, the researchers measured relative pupil constriction (amplitude and latency of constriction) in response to the white flash. The pupillary light reflex is thought to index activity of the cholinergic system, which is proposed to be disrupted in ASD. The original data comparing high risk and normal risk 10 month old infants was presented in Nystrom 2015 and identified a large difference between the two groups. Specifically, infants at higher risk for developing ASD exhibited greater relative pupil constriction that occurred earlier than normal risk controls. The data presented here includes the 24 month ADOS outcome data on these infants, further splitting the children into three groups: ASD, no-ASD high risk, and TD. For the most part methodologically, this study has been reviewed and published already (Nystrom et al., 2015). Thus, the majority of this review will focus on what is new to this paper compared to Nystrom et. al., 2015. Nevertheless, I suggest that the authors give more expository information regarding the original experimental design, so that readers unfamiliar with Nystrom et al., 2015 have an easier time orienting themselves to this study . Below are major and minor points that the authors should address.

Specific points:

- The authors used the Calibrated Severity Score from the ADOS to designate groups. Did the children receive a clinical DX of autism that was the same as what was predicted by the CSS? Alternatively, did the clinician judge that they were too young to be assigned a stable diagnosis of ASD? Either way it would helpful to know whether the children received a diagnosis.
- The CSS ranges from 1-10. What was the direct relationship between the CSS and the PLR? How did this relationship compare to the group based statistics reported here?
- What was the relationship between the amplitude and the latency of the PLR response? It looks

like they're sharing variance that is related to the effect distinguishing groups. Would the 1st principal component of the two variables predict group membership better?

- The Mullen scales did not distinguish between groups, but they might share some meaningful relationship with the PLR. Can the authors present those data? The small sample size of the ASD group might mean that the tests don't reach significance between groups, but there might still be a relationship with a meaningful effect size.
- The typical male/female ratio of individuals with ASD was reversed in this sample. Across all individuals what was the relationship with sex and PLR. Could this explain the differences seen across groups? Can the authors plot the PLR response by sex?
- When the children returned for the follow up visit, did they participate in the PLR experiment?
- Can we see the data for the 12 children who did not return for the follow up visit? Given the small ASD group in this sample it would be helpful to see the full distribution of the data collected and where data from those children fell.

Overall, this is an impressive paper that makes a strong case for the utility of the PLR as an early biomarker for ASD status. However, there are several concerns (1) the sample size for the ASD positive group is small and does not exhibit the typical sex ratio seen in ASD; (2) it is unclear whether these children actually received a diagnosis of ASD; (3) it looks like there should be a strong continuous relationship between the PLR and clinical characterization. The authors should present these data as well.

Responses to Reviewers' Comments:

Reviewers' comments:

Reviewer #1 (Remarks to the Author):

The manuscript reports stronger pupillary light reflex (PLR) in infants at-risk for autism relative to those at-risk who do not develop autism or typically developing controls. They report eye tracking data that has been recently published (Molecular Autism 2015), but the current manuscript presents a re-analysis of PLR data against diagnostic outcomes at 2- years of age. Previous studies have reported differences in visual processing in infants later diagnosed with ASD, e.g., Gliga et al 2014 Curr Bio; Shic et al BioPscyh). Yet, this is the first clear demonstration that a fairly low-level visual reflex is associated with a later diagnosis of autism.

The authors base their hypothesis on prior studies that suggest attenuated PLR in older children and adults with ASD (where their prior work suggests the opposite pattern earlier in life). The manuscript and the conclusions would be much stronger if this attenuation is directly observed in the same sample over development (between 10 months and 2-4 years). This is especially important given the small samples size of diagnosed children. Therefore, the conclusions would be bolstered by the inclusion of PLRdata on the same infants at 2-3 years. My concern here is that overt behavioural risk

signs of autism (affecting social interaction) emerge rather clearly around 12-months of age, which is only two months after the age that a stronger PLR is reported.

RESPONSE2: *see response1*

Also, further follow up to 3-years to examine stability of diagnosis would be needed, given that only a provisional diagnosis can be ascertained at 2-years.

RESPONSE3: *we now use DSM-5 ASD diagnosis at 3 years as the outcome variable.*

Methods used to calculate the PLR appear sound. However, my knowledge of the Tobii system with infants is that raw data is often contaminated by a number of artefacts. It would be great if the authors can provide further demonstration that these artefacts cannot account for the observed group differences, e.g., analysis of baseline data to show it does not differ in the three groups.

RESPONSE4: *We have visually inspected all trials and excluded trials where there were signs of artifacts (as explained in the new methods description). We have also analyzed baseline differences and compared the amount of missing data from each group, as artifacts often cause missing data. These analyses are found in the SOM, and show that there are no differences in these factors and that they are unlikely to affect our main results.*

In terms of interpretation, I agree with the author's conclusion that the observed differences may reflect genetic liability. The current manuscript brings us a few steps closer to understanding potential genetic and cellular mechanisms, given that the PLR is biologically well characterized.

In contrast, I don't agree with the authors' occlusion that they have identified a "developmentally sensitive predictive biomarker of ASD". I think this is a misinterpretation of what a biomarkers is and its potential clinical utility. It's important to note, however, that Nature has set a precedent for

publishing conclusions about eye tracking in infants that have much bigger news than scientific or clinical value (Klin & Jones 2014). Therefore, I hope that the authors can maintain their focus on advancing scientific discovery without the pressure to prematurely suggest clinical utility where there is none.

RESPONSE 5: *this point is well taken, and we have moderated the language accordingly throughout the text.*

Reviewer #2 (Remarks to the Author):

This study used the high-risk infant sibling design to investigate whether differences in the pupillary light reflex (PLR) may be an early risk or diagnostic marker for Autism Spectrum Disorder (ASD). It builds on a previously published study (Nystroem et al, 2015, Mol Autism) by the same group, which reported that 10-month old infants with high familial risk for ASD had a stronger PLR than low-risk infants.

ASD can currently only be reliably clinically diagnosed from around the age of 24 months onwards. Therefore, the current report goes crucially beyond the previous study by now comparing high-risk infants who did vs. did not develop ASD on the PLR.

I found the manuscript to be well written and the study design very clear. I agree that the PLR is a very interesting candidate biomarker for ASD for the reasons that were given.

However, I also had several queries/ concerns:

1. The sample size of the HR-infants who developed ASD is quite small (N=6). This should be considered in the context of the considerable costs of the HR infant design, both due to its longitudinal nature and because only (very roughly) 1 in 5 children that are originally enrolled

develop ASD. Therefore, I would treat the results as potentially very interesting but in my opinion at this point they do not provide “strong evidence” for the value of the PLR as a risk/ diagnostic marker.

RESPONSE 6: *this is a highly valid point also raised by the editor and other reviewers, see Response1 on how we handled this by adding more data from an independent lab. Also, focusing on 3 year diagnostic outcome increased the N of the ASD group in the original sample from Sweden. The study now includes n = 29 HR siblings with an ASD diagnosis at 3 years of age.*

2. As in many studies, the authors seem to conclude from their finding of a significant group difference that the measure could be promising risk or diagnostic biomarker. However, the aim of a risk or diagnostic biomarker is to predict, for a given individual, whether or not they likely develop or have ASD. This can't be done based on mean group differences but requires at least an assessment of the accuracy of the measure (i.e., its sensitivity/ specificity). Individual values are nicely plotted in Fig 1 c for relative constriction and latency. But from that it appears as if there was some overlap in scores between the three groups. For example, on relative constriction three out of the six 'ASD positive' children' had scores within 2 SDs of the TD range (but above the TD mean). Conversely, three of the TD children seem to have scores within or above the 'ASD positive' mean. More information would be useful that gives some estimate of how sensitive/ specific would be the PLR in predicting if a child develops or has ASD?

RESPONSE 7: *this point echoes Reviewer1's comment, and we now are careful not to imply that we have identified a predictive diagnostic biomarker. This was indeed an overstatement in the previous version, as we do not think the current measures can be said to have these characteristics.*

3. The authors mention in the introduction that while they observed (on average) stronger PLRs in high-risk infants, other previous studies with children and adults with ASD found weaker PLRs. They

argue that therefore the PLR may not only predict ASD but also “reveal neural processes that are on a highly deviant developmental trajectory”. Again, I understand their point and agree that this would be an interesting possibility. However, the current study is not able to support/ speak to this; indeed there remains a possibility that differences in these study findings could be due to other factors that may be important to understand in order to assess the potential clinical biomarker utility of the PLR.

RESPONSE 8: *See our RESPONSE 1,we now provide new data supporting this notion.*

Reviewer #3 (Remarks to the Author):

The authors present follow up data on a study of high-risk infants who participated in a pupillary light reflex (PLR) experiment at 10 months of age. Between trials of point light displays, infants saw a 75ms white flash preceded and followed by a black screen. Using a video based eye-tracker, the researchers measured relative pupil constriction (amplitude and latency of constriction) in response to the white flash. The pupillary light reflex is thought to index activity of the cholinergic system, which is proposed to be disrupted in ASD. The original data comparing high risk and normal risk 10 month old infants was presented in Nystrom 2015 and identified a large difference between the two groups. Specifically, infants at higher risk for developing ASD exhibited greater relative pupil constriction that occurred earlier than normal risk controls. The data presented here includes the 24 month ADOS outcome data on these infants, further splitting the children into three groups: ASD, no-ASD high risk, and TD.

For the most part methodologically, this study has been reviewed and published already (Nystrom et al., 2015). Thus, the majority of this review will focus on what is new to this paper compared to Nystrom et. al., 2015. Nevertheless, I suggest that the authors give more expository information regarding the original experimental design, so that readers unfamiliar with Nystrom et al., 2015 have an easier time orienting themselves to this study .

RESPONSE 9: *We have carefully gone through the methods to make sure it can be understood without reading the previous paper. Also, the additional independent dataset moves this paper well beyond the original Nyström 2015 paper.*

Below are major and minor points that the authors should address.

Specific points:

- The authors used the Calibrated Severity Score from the ADOS to designate groups. Did the children receive a clinical DX of autism that was the same as what was predicted by the CSS? Alternatively, did the clinician judge that they were too young to be assigned a stable diagnosis of ASD? Either way it would be helpful to know whether the children received a diagnosis.

RESPONSE 10: *as noted above, the outcome variable is now DSM-5 ASD diagnosis at 3 years of age.*

- The CSS ranges from 1-10. What was the direct relationship between the CSS and the PLR? How did this relationship compare to the group based statistics reported here?

RESPONSE 11: *no longer applicable as we have changed to 3 year ASD diagnosis rather than ADOS as outcome variable. However, we have now added dimensional analyses from both ADI-R and ADOS2 that show the relationship between these measures and the PLR parameters.*

- What was the relationship between the amplitude and the latency of the PLR response? It looks like they're sharing variance that is related to the effect distinguishing groups. Would the 1st principal component of the two variables predict group membership better?

RESPONSE 12: *This is a good question, and the use of PCA could potentially improve the group predictions. However, because this analytic approach would mix two processes with different error*

sources it could be difficult to interpret the result in terms of underlying processes, and we have chosen not to include this analysis.

- The Mullen scales did not distinguish between groups, but they might share some meaningful relationship with the PLR. Can the authors present those data? The small sample size of the ASD group might mean that the tests don't reach significance between groups, but there might still be a relationship with a meaningful effect size.

RESPONSE 13: *This is an interesting idea and we have now added this analysis in the Supplementary Materials. The analysis and scatter plot show that there is no relation between the 10 month Mullen data with the PLR relative constriction.*

- The typical male/female ratio of individuals with ASD was reversed in this sample. Across all individuals what was the relationship with sex and PLR. Could this explain the differences seen across groups? Can the authors plot the PLR response by sex?

RESPONSE 14: *We have now added a GLM with gender as an additional fixed factor. The results are found in the SOM, and show no differences between girls and boys. With this analysis at hand it felt unmotivated to include a plot of the PLR response by sex, but we can add it if requested. Also, the current study, with data from two sites, has a more "traditional" male/female ratio than the previous version (HR-ASD n=29, 7 girls; HR-no-ASD n=118, 62 girls; TD n=40, 20 girls).*

- When the children returned for the follow up visit, did they participate in the PLR experiment?

RESPONSE 15: *The PLR assessment was conducted longitudinally in the infant assessments in both sites (10-18m in the EASE sample; 9-15m in BASIS sample), but not at 2 or 3 years.*

- Can we see the data for the 12 children who did not return for the follow up visit? Given the small ASD group in this sample it would be helpful to see the full distribution of the data collected and where data from those children fell.

RESPONSE 16: *no longer applicable given the substantially larger N in all groups, and as the outcome data and overall design (two sites) has changed.*

Overall, this is an impressive paper that makes a strong case for the utility of the PLR as an early biomarker for ASD status. However, there are several concerns (1) the sample size for the ASD positive group is small and does not exhibit the typical sex ratio seen in ASD; (2) it is unclear whether these children actually received a diagnosis of ASD; (3) it looks like there should be a strong continuous relationship between the PLR and clinical characterization. The authors should present these data as well.

RESPONSE 17: *the current study consists of 29 children with ASD diagnoses and the current gender ratio is 7:22 (i.e. 7 girls, 22 boys, or 3.14 times more boys). We now report relations between the PLR parameters and continuous measures from ADI-R and ADOS-2. The results show significant correlations in many of the dimensional measures.*

Reviewers' Comments:

Reviewer #1 (Remarks to the Author):

The revised manuscript reports stronger pupillary light reflex (PLR) in infants at-risk for autism relative to those who do not develop autism or typically developing infants with no family history of autism. The revision includes a much larger group of infants, substantially larger than most publications in this area to date. The authors achieved this by collaborating with research another group. The sample now includes a sizable group diagnosed with autism as toddlers. As such, issues raised in the previous review have been fully addressed. The article will make a very strong contribution, in part because of the impressive sample size and the novelty of the findings. More importantly, the results call in question dominant theoretical positions in the field (including one published in Nature) that problems in social engagement are the primary risk manifestation in autism. I am certain that this manuscript will make a significant impact on the field and generate further dialogue.

By way of slight improvement to the current version, I suggest analyzing ADOS and AD scores only for the high risk group since these measures were not validated for the control group and the data appears skewed. It's likely that high symptom scores in few toddlers reflect alternative behaviours, e.g., compliance with experimenter or attentional difficulty.

Reviewer #2 (Remarks to the Author):

Overall, I was impressed by the amount of revisions undertaken by the authors, and the now significantly increased sample size (which was achieved by including a second site). This clearly further improved the overall impact of the study. I only had the following remaining comments/queries:

The relationship between relative pupil constriction during infancy and RRBs at 3 years was rather a 'trend' than significant, and should probably be reported as such ($n=179$, $r=0.140$, $P=0.062$). Further, this relationship became non-significant when the TD children were excluded. ADOS RRB $n=144$, $r=0.136$, $P=0.103$.

I would therefore suggest revising the relevant sentences in the results section and the sentence "The results of this study show that that the magnitude of the pupillary light reflex constriction in infancy is associated with ASD diagnosis at three years of age, and that it captures neural

processes related to both social functioning and behavioral flexibility (RRB).” to omit reference to the RRB.

While I agree that a relationship with RRBs may be expected because this diagnostic category includes now sensory anomalies, one factor in this lower-than-expected association may be that the ADOS-RRB score was not designed to capture sensory atypicalities.

It was unclear to me why the authors decided to use a difference score to test group differences in the longitudinal trajectories, rather than a mixed ANOVA with the factors group x assessment point. This should be justified. Supplementary Figure 1 illustrating the differences in the developmental trajectories should probably also include the HR-no ASD group.

Minor comments:

Abstract:

Consider changing ‘cure’ for ‘effective treatment’

Should insert “on average” in the following sentence: Here, we show that the relative constriction of the pupillary 40 light reflex is “on average” larger in 9-10 month-olds high risk infant siblings....

Reviewer #3 (Remarks to the Author):

The authors have addressed my concerns from the earlier manuscript and made substantial changes.

However, the authors have opted to use an unusual standardization approach for the pupil light response, dividing by the average of the TD group at each site. This approach appears to be effective, but it would be helpful to see more detail in the algorithm and presentation of site differences before normalization.

Responses to Reviewers' Comments:

Reviewer #1 (Remarks to the Author):

The revised manuscript reports stronger pupillary light reflex (PLR) in infants at-risk for autism relative to those who do not develop autism or typically developing infants with no family history of autism. The revision includes a much larger group of infants, substantially larger than most publications in this area to date. The authors achieved this by collaborating with research another group. The sample now includes a sizable group diagnosed with autism as toddlers. As such, issues raised in the previous review have been fully addressed. The article will make a very strong contribution, in part because of the impressive sample size and the novelty of the findings. More importantly, the results call in question dominant theoretical positions in the field (including one published in Nature) that problems in social engagement are the primary risk manifestation in autism. I am certain that this manuscript will make a significant impact on the field and generate further dialogue.

By way of slight improvement to the current version, I suggest analyzing ADOS and AD scores only for the high risk group since these measures were not validated for the control group and the data appears skewed. It's likely that high symptom scores in few toddlers reflect alternative behaviours, e.g., compliance with experimenter or attentional difficulty.

ANSWER TO REVIEWER #1: This is a good point, and we have changed the manuscript accordingly, by changing the text to focus to the HR only analysis (and only plot the HR group in figure 2).

Reviewer #2 (Remarks to the Author):

Overall, I was impressed by the amount of revisions undertaken by the authors, and the now significantly increased sample size (which was achieved by including a second site). This clearly further improved the overall impact of the study. I only had the following remaining comments/ queries:

The relationship between relative pupil constriction during infancy and RRBs at 3 years was rather a 'trend' than significant, and should probably be reported as such ($n=179$, $r=0.140$, $P=0.062$). Further, this relationship became non-significant when the TD children were excluded. ADOS RRB $n=144$, $r=0.136$, $P=0.103$.

I would therefore suggest revising the relevant sentences in the results section and the sentence "The results of this study show that that the magnitude of the pupillary light reflex constriction in infancy is associated with ASD diagnosis at three years of age, and that it captures neural processes related to both social functioning and behavioral flexibility (RRB)." to omit reference to the RRB.

While I agree that a relationship with RRBs may be expected because this diagnostic category includes now sensory anomalies, one factor in this lower-than-expected association may be that the ADOS-RRB score was not designed to capture sensory atypicalities.

ANSWER TO REVIEWER #2: We agree to these well-taken points, and have revised the manuscript accordingly. In response also to reviewer #1 we have now rewritten the RRB section and report the $P=0.062$ as a trend.

It was unclear to me why the authors decided to use a difference score to test group differences in the longitudinal trajectories, rather than a mixed ANOVA with the factors group x assessment point. This should be justified. Supplementary Figure 1 illustrating the differences in the developmental trajectories should probably also include the HR-no ASD group.

ANSWER: Statistically, the reported result is the exactly the same as one would get from testing an interaction effect with ANOVA. So this is just a matter of taste – we think the differences score most effectively addresses the matter of interest: (group differences in) change over time. To make this clear, we note in the manuscript that the statistics are identical to what one would get from testing an interaction effect in an ANOVA.

Minor comments:

Abstract:

Consider changing ‘cure’ for ‘effective treatment’

ANSWER: When shortening the abstract to <150 words this sentence was deleted, so this comment is no longer applicable.

Should insert “on average” in the following sentence: Here, we show that the relative constriction of the pupillary 40 light reflex is “on average” larger in 9-10 month-olds high risk infant siblings....

ANSWER: Done.

Reviewer #3 (Remarks to the Author):

The authors have addressed my concerns from the earlier manuscript and made substantial changes.

However, the authors have opted to use an unusual standardization approach for the pupil light response, dividing by the average of the TD group at each site. This approach appears to be effective, but it would be helpful to see more detail in the algorithm and presentation of site differences before normalization.

ANSWER TO REVIEWER #3: We have now added more information how we transformed the data, and have also made a reference to the supplementary materials where site differences before normalization are presented.